# Three-Channel Metasurfaces for Multi-Wavelength Holography and Nanoprinting

**DOI:** 10.3390/nano13010183

**Published:** 2022-12-31

**Authors:** Zuyu Li, Yuhang Zhang, Jiadong Yuan, Yuhan Hong, Hongzhan Liu, Jianping Guo, Qiaofeng Dai, Zhongchao Wei

**Affiliations:** Guangdong Provincial Key Laboratory of Nanophotonic Functional Materials and Devices, School of Information and Optoelectronic Science and Engineering, South China Normal University, Guangzhou 510006, China

**Keywords:** metasurfaces, holographic image, nanoprinting image, multi-wavelength control

## Abstract

Metasurfaces, employed to simultaneously generate nanoprinting and holographic images, have been extensively explored recently. Among them, multi-wavelength multiplexing in a single metasurface is often accompanied by dispersion and crosstalk, which hinder the display of multicolor patterns. Here, we propose an efficient phase method to decouple the wavelength and realize a three-channel display operating at different wavelengths. Holographic images appear in the far field with the illumination of two different circularly polarized lights while a nanoprinting image is reconstructed by inserting an orthogonal optical path with the illumination of linear polarization light. The proposed metasurface is only composed of four types of unit cells, which significantly decreases the complexity of fabrication and improves the information capacity. Benefiting from its different decoding strategies and capability of multi-wavelength control, this approach may develop broad applications in information encryption, security, and color display.

## 1. Introduction

Conventional single-channel information encoding usually records one-dimension information but lacks a complete description of the object, which greatly restricts the application in practice. As a developing technology with high information density, multi-channel information encoding is widely applied to many fields, such as AR, VR, and deep learning [1,2,3,4]. Taking VR as an example, it integrates three-dimensional data of a practical environment with computer simulation to get a more realistic experience compared with traditional video. With the rapid development of information technology, the demand for information in multiple channels is getting larger increasingly. Integrating multidimensional information into a smaller device has become an urgent need to improve information density.

Recently, owing to the unprecedented ability to modulate electromagnetic waves, metasurfaces can precisely manipulate the phase, amplitude, and polarization of the incident light with subwavelength dimensions [5,6,7,8,9], which pave a new path to encode and display multi-channel images [10,11,12,13,14,15,16,17,18]. Among them, nanoprinting and holographic images are emerging research fields. Nanoprinting appears in the near-field, which is modulated by the amplitude of light [19,20], while the holographic image appears in the far-field, which is usually modulated by the phase [21,22,23]. The key to simultaneously displaying images in both fields is to manipulate the phase and amplitude on the metasurfaces. Utilizing this approach, various metasurfaces merging holography and nanoprinting have been proposed to generate images in both fields [8,24,25,26]. Due to the dispersion effect and the limited degrees of freedom, the nanoprinting and holographic images often accompany some cross-talk between each wavelength [27,28,29,30]. A common scheme to avoid cross-talk is to generate a specific spectral response by varying the dimensions of the nanostructure based on structural color. For example, the structural color is employed to generate colorful nanoprinting images in the near-field and the orientation angles of the nanostructure modulate the geometric phase to produce holographic images [31]. Some schemes achieve multi-wavelength display by arranging two or more nanostructures as a supercell. The amplitude of each wavelength can be meticulously modulated with the different orientation angles [32]. In addition, an emerging solution is to add specific dielectric into metasurface such as oil and dimethyl sulfoxide (DMSO). The refractive index or resonant mode are varied when adding a specific dielectric, which stimulates more spectral response [33,34]. However, these metasurfaces require a large number of nanostructures with varied dimensions to satisfy the phase and amplitude at different wavelengths. For instance, structural color requires plenty of dimensions to generate various spectral responses, which increases the difficulty of fabrication. Moreover, the complex structures and application conditions such as arranging supercells and adding specific dielectrics hinder the practical application of multi-wavelength displays.

Here, we demonstrate a three-channel metasurface to simultaneously display a nanoprinting image and two-wavelength holographic images with only four dimensions of nanostructures, as shown in Figure 1. Channel 1 is a nanoprinting image obtained by a linear polarization (LP) light that successively passes the metasurface and an analyzer. Channel 2 and channel 3 are holographic images obtained by circularly polarized (CP) light at two different wavelengths. In our design, the required amplitude in the near-field is derived by the orientation angles of nanostructures based on Malus law and the phase is controlled by orientation degeneracy and propagation phase. By the ingenious combination of geometric phase and propagation phase, as for each pixel, we can get a specific table with 16 phase groups covering 2π phase at two wavelengths [35]. The phase group that simultaneously satisfies both holograms at the two wavelengths can be chosen and holographic images in channel 2 and channel 3 are encoded into the metasurface. Due to the full control of amplitude and phase at the three wavelengths, the cross-talk caused by the spectral response can be largely avoided and the required dimension types of nanostructures can be significantly reduced. More interestingly, different spin polarization lights can be employed to be the incident light to display holographic images in another strategy, which increases the flexibility of the design. Compared with previous work, our approach provides a new idea to achieve multi-wavelength display with a simpler structure and may be widely applied to optical encryption, color holography, VR, etc.

## 2. Principle and Design

First, the nanoprinting image in the near-field requires independent intensity modulation. Governed by Malus law, when the nanostructure acts as a nano-half-waveplate, the amplitude can be precisely manipulated by rotating angles. When an LP light passes through the nanostructure and the analyzer is placed in the orthogonal direction, the output light intensity can be expressed by the formula [36]:(1)$$I=I_{0}{\left(\frac{T_{L}-Ts}{2}\right)}^{2}\sin^{2}(2\theta )$$
where *I*_0_ represents the intensity of incident light and *T_L_* and *T_S_* are the complex transmission coefficients of the nanostructure along the long axes and short axes respectively, *θ* is the orientation angle of the nanostructure. When the nanostructure acts like a nano-half-waveplate (A = 1, B = −1), the output intensity can be expressed as $I=I_{0}\sin^{2}\left(2\theta \right)$. So, the intensity of output light can be continuously modulated by configuring the orientation of each nanostructure.

Interestingly, the orientation degeneracy provides a new degree of freedom to modulate the cross-polarized phase. As illustrated in Figure 2, four orientation angles in the same nanostructure can generate an equal beam intensity, which can be employed to modulate phase. In the principle of geometric phase modulation, under LCP light, the complex amplitude of output light can be written as this formula [37]:(2)$$E_{OUT}=\frac{T_{L}+T_{S}}{2}\left(\begin{array}{l} 1\\ i \end{array}\right)+\frac{T_{L}-T_{S}}{2}e^{i2\theta }\left(\begin{array}{l} 1\\ -i \end{array}\right)$$
where *T_L_* and *T_S_* represent the complex transmittance coefficients along the long axis and short axis respectively. From Equation (2), the cross-polarized component takes 2*θ* phase delay under LCP light. Therefore, each nanostructure can allocate a four-level geometric phase to encode the hologram.

The two holograms at two different wavelengths need to be independently modulated by phase. Normally, for the purpose of simultaneously realizing m-level phase control at two wavelengths with the propagation phase, the total of required nanostructures would be a large value of m^2^. To avoid a great number of required dimensions, an improved method is proposed. The four candidate orientation angles are derived by using Malus law, which provides additional four-level phase modulation to the cross-polarized component of CP light. In order to obtain more phase combinations covering 2π at both wavelengths, we screen out the dimensions with different propagation phases at the two wavelengths and list each phase with a different rotation angle and dimension. Therefore, 16 phase groups containing different phase responses at the two wavelengths are derived. Assuming that the intensity of a near-field pixel is 0.5, the candidate orientation angles are 22.5°, 67.5°, 112.5°, and 157.5°, respectively. The propagation phase differences between the two wavelengths of nanostructures are set to π/2, π, 3π/2, and 2π, respectively. As shown in Table 1, 16 phase groups are listed in the case of 0.5 intensity. By comparing two phase profiles at the two wavelengths, the phase group closest to both target phases can be chosen. Therefore, the corresponding dimension and orientation angle is selected. In this way, the number of required nanostructures is reduced to m. It is noted that the four candidate angles are obtained by the intensity of each pixel in the near-field. If another intensity of a pixel in near-field requires to be encoded to 1/4, according to Malus law, the candidate orientation angles would be 15°, 75°, 105°, and 165°, respectively and the corresponding phase groups should be changed accordingly. Figure 3a,b show the candidate phase distribution respectively in 1/2 and 1/4 intensity, from which we can see that the phase distribution in 1/2 intensity is more uniform. Therefore, when the grayscale of the nanoprinting image is close to 1/2, the phase at both wavelengths will be better modulated.

Next, the target phase profile of holography is calculated by Gerchberg–Saxton (GS)algorithm. GS is a simple, widely applied algorithm that can optimize holograms with high quality. In order to apply it to the case of a large angle exit, Rayleigh Sommerfeld diffraction (RS) is used to calculate the diffraction result. The formula of diffraction can be given by [38]:(3)$$U(x_{0},y_{0})=\frac{1}{i\lambda }\iint U(x,y)\cos<n,r>\frac{\exp(ikr)}{r}dxdy$$
where *U*(*x*_0_, *y*_0_) and *U*(*x*, *y*) represent the complex amplitude distribution of electric fields on the holographic plane and metasurface, respectively. cos <*n*, *r*> is the inclination factor, $r=\sqrt{{\left(x-x_{0}\right)}^{2}-{\left(y-y_{0}\right)}^{2}+z^{2}}$ and z is the propagation distance. With the GS algorithm optimization, two individual phase profiles for holograms at two different wavelengths can be derived.

The schematic diagram of designing the proposed metasurfaces is illustrated in Figure 4. First, the intensity in the near-field is calculated by Malus law. Matching the target images in the curve and then the four candidate orientation angles are obtained. By a combination of different propagation phases at both wavelengths, 16 phase groups covering 2π at both wavelengths can be listed. Different intensity of a pixel in the nanoprinting image corresponds to different phase groups. The phase-only hologram at both wavelengths is obtained by the GS algorithm. Screen out the phase group closest to the two holograms and then the corresponding dimension and rotation angle can be selected. Through the above process, the two different holographic images in the far-field with different wavelengths of CP light and a nanoprinting image at a wavelength of LP light are encoded into the metasurfaces.

## 3. Results and Discussion

### 3.1. Unit-Cell Design

TiO_2_ and SiO_2_ have the advantage of high transmittance in visible and near-infrared regions, which can improve holographic diffraction efficiency. Therefore, the TiO_2_ nanobrick is employed as the nanostructure, and SiO_2_ is employed as the planar substrate, as shown in Figure 5a. We utilize FDTD Solutions (Lumerical Inc., Vancouver, BC, Canada) to scan and select suitable dimensions. Periodic boundary conditions are applied in both the x and y directions and the perfectly matched layer (PML) boundary condition is applied in the z-direction. In order to cover the 2π phase difference between the two wavelengths, the height of each nanobrick is identically set to be 1000 nm and the cell size is set to be 400 × 400 nm. The incident light is set to be an LCP light. Next, we scan the dimension of the nanobrick to obtain the propagation phases at each wavelength. In the simulation, the length of the nanobricks ranges from 150 nm to 350 nm, and the width ranges from 50 nm to 250 nm with a step of 10 nm. The phase differences between two wavelengths are determined by the dimensions of nanostructures which can stimulate different resonance modes. To cover 2π phase difference at both wavelengths, the differences between two wavelengths cannot be too small. So, we carefully choose four dimensions of nanobricks operating at 500 nm and 680 nm. The details of dimension parameters are listed in Figure 5b. Figure 5c–f plots the propagation phase difference ranging from 0 to 3π/2 between 500 nm and 680 nm of each dimension.

Furthermore, we investigate the polarization conversion rate (PCR) in each dimension at the range of 500 to 700 nm, as exhibited in Figure 6a. Based on Malus law, the nanostructure is required to be a half-wave-plate (HWP) to continuously modulate amplitude. Uneven PCR of each nanostructure will cause inaccurate encoding of the nanoprinting grayscale. Therefore, the wavelength corresponding to the highest PCR in each dimension will be chosen. Here, 596 nm is chosen to be the operation wavelength for nanoprinting, which is the suitable wavelength with high PCR among the nanobricks. Figure 6b plots the PCR of each dimension at the three wavelengths. The PCR of each dimension at 596 nm is higher than 90%. Furthermore, the PCR at 680 nm is higher than the PCR at 500 nm, so the efficiency of the holographic image at 680 nm will be higher.

### 3.2. Demonstration and Discussion of Three-Channel Metasurfaces for Multi-Wavelength Holography and Nanoprinting

To verify our design, we design two samples of metasurfaces for the three-channel display of two different holographic images and one nanoprinting image. The metasurfaces consist of 100 × 100 nanobricks and the whole size of metasurfaces is 40 μm × 40 μm. The designed distance of metasurfaces to the holographic image is 30 μm. As for holographic images, we employ two LCP lights to illuminate the metasurfaces, respectively, at 500 nm and 680 nm. As for the nanoprinting image, the LP light at 596 nm is placed in front of metasurfaces and an orthogonal analyzer.

For the target images of holography, due to the discretization of the phase, displaying binary grayscale images are more easily. As shown in Figure 7, the holographic images at 680 nm and 500 nm are “A” and “B”, respectively, in the distance of 30 μm and the nanoprinting image in sample 1 is a “snowman”. Sample 2 shows a “rabbit” in the near-field and the two holographic images are “clover” and “banana”. When the metasurface is illuminated by an LCP light at 500 nm and 680 nm, two independent holographic images are displayed at 30 μm respectively with high-definition. When the metasurface is inserted into an orthogonal optical path and illuminated by an LP light at 596 nm, a nanoprinting image determined by light intensity distribution can be clearly seen in the near-field.

Further, we investigate the holographic efficiency of the two holographic images. The measured holographic image efficiency can be defined as the ratio between the light intensity of holographic imaging and input intensity. At the designed wavelength of 680 nm, the holographic image efficiency in sample 1 and sample 2 are 48.14% and 48.67% respectively. At 500 nm, the efficiency in sample 1 and sample 2 are 24.9% and 23.83%. It attributes that the PCR of each nanobrick dimension at 500 nm is lower than the latter which is mentioned in Figure 6b. Compared with other metasurfaces, our achieved holographic image efficiency is reasonable [31]. Actually, to simultaneously operate at multiple wavelengths, the diffraction efficiency of multi-wavelength metasurfaces is usually lower than the efficiency of the single-wavelength metasurface. The reduced efficiency of our metasurface mainly comes from the nanobricks with low PCR at holographic wavelength. Due to the need to cover the 2π phase difference between 680 nm and 500 nm, it is difficult to derive the dimensions simultaneously satisfying high PCR at both wavelengths. By screening more dimensions, the suitable wavelength with higher PCR can be chosen to adapt to application needs.

To explore the broadband characteristic of the holographic images in the far-field, we employ four LCP lights near the operation wavelengths to illuminate the metasurface and observe the holographic images in the designed distance. As shown in Figure 8, the holographic images work well in a wide band. According to the diagram plotted in Figure 5c–f, there is no abrupt change of the propagation phase near the designed wavelength, which ensures the phase approximately matching for holography. Furthermore, when the wavelength deviates more from the designed wavelength, the holographic image will emerge with larger geometric distortion. This phenomenon can be accounted for by the phase mismatch based on diffraction theory. Interestingly, when the wavelength of input light changes from 500 nm to 700 nm, there will be a gradient effect from one image to another.

In addition, the four candidate angles can allocate two groups to encode a binary image beside the three channels. According to Malus law, if the Jones vector of the incident light is ${\left[\cos\alpha_{1},\sin\alpha_{1}\right]}^{T}$ and the optic analyzer is placed behind the metasurface with direction *α*_2_, the intensity of output light can be expressed as followed [39]:(4)$$I={\left|\frac{T_{L}-T_{S}}{2}\cos(2\theta -\alpha_{1}-\alpha_{2})+\frac{T_{L}+T_{S}}{2}\cos(\alpha_{1}-\alpha_{2})\right|}^{2}$$
where *θ* represents the orientation angle of the nanostructure. When the metasurface operates at 596 nm, the coefficient $\frac{T_{L}+T_{S}}{2}$ can be ignored due to the half-wave plate of each nanostructure at 596 nm. Consider the Equation (4) in this case: *α*_1_ = 22.5° and *α*_2_ = 112.5°, the intensity of the output light can be expressed as
(5)$$I_{1}=\frac{T_{L}-T_{S}}{2}\sin^{2}(2\theta -\frac{\pi }{4})$$

Therefore, it provides a new freedom degree to encode a binary nanoprinting image. The working principle is shown in Figure 9a. The continuous grayscale image is modulated by the orientation angle and the binary image is modulated by the orientation degeneracy based on Equation (5). Then the table consisting of 4 × 2 phase groups can be listed to modulate the dual-wavelength holography. Figure 9b shows the simulation results of this four channels scheme, where a tree can be clearly seen in the case of *α*_1_ = 0°, *α*_2_ = 90°, and “Y” can be seen in the case of *α*_1_ = 22.5°, *α*_2_ = 112.5° in the near-field. By illuminating the metasurface with LCP light at 680 nm and 500 nm, the holographic images (“G” and “F”) are displayed, respectively, in the far-field. Due to the reduction of the phase groups, the holographic images will undergo some deterioration.

Since the four candidate angles governed by Malus law cover 2π phase, the proposed method can be extended to encode different spin polarization of CP light. In the principle of geometric phase modulation, under RCP light, the complex amplitude of output light can be written as this formula [37]:(6)$$E_{OUT}=\frac{T_{L}+T_{S}}{2}\left(\begin{array}{l} 1\\ -i \end{array}\right)+\frac{T_{L}-T_{S}}{2}e^{-i2\theta }\left(\begin{array}{l} 1\\ i \end{array}\right)$$

Equations (2) and (4) indicate that the geometric phase modulations of LCP light and RCP light are completely opposite. When the incidence light at 500 nm is LCP light while the incidence light at 680 nm is RCP light, the phase groups of 1/2 intensity in near-field can be listed in Table 2. At the two wavelengths, the two CP lights with different rotation directions have completely opposite geometric phase modulation, which is different from Table 1. Similarly, by choosing the appropriate phase group in Table 2, the two independent holographic images can be simultaneously encoded into the metasurface. Figure 10 shows the two holographic images (“C” and “D”) under RCP and LCP in the case of the “tiger” nanoprinting image. The results indicate that the proposed method artificially controls the phase according to the polarization of the incident light with the same kinds of nanostructures, which broadens the selection of incidence light.

The proposed metasurfaces for three wavelengths holography and nanoprinting provide a simple approach for multi-wavelength control in near-field and far-field. There are several distinct advantages in our design over previous schemes. First of all, the metasurface only requires four kinds of nanobrick dimensions, which significantly lightens the difficulty of metasurfaces fabrication. Meanwhile, it simplifies the complicated process for screening dimensions to obtain suitable propagation phases at both wavelengths. Second, the broadband characteristic in the far-field improves the robustness and practicability of the three-channel metasurfaces. Third, it offers a way to encode four channels of nanoprinting and holographic images, which additionally encode a binary nanoprinting image by orientation degeneracy. At last, the three images need to be encoded by three different wavelengths and the spin polarization of incidence light of holography can be artificially designed. Specifically, to completely decode the three channels, the three kinds of operation wavelengths and their corresponding polarization states should be given in advance, which constitute three different secret keys. With the help of the special service condition, our approach paves a new path for image encryption and significantly increases the security of image display.

## 4. Conclusions

In general, we propose and demonstrate a three-channel metasurface for multi-wavelength holography and nanoprinting merely with four dimensions of the structure. By combining amplitude modulation and phase modulation based on Malus law and phase groups, we achieve three independent image displays in the near-field and far-field at three different wavelengths. In our design, only four types of nanostructures are required to compose the metasurface, which significantly simplifies the structure and reduces the difficulty of fabrication without reducing information density. With the advantages of high information density, high security, and strong robustness for three-image display, the proposed metasurface will bring great convenience for information storage, integration, and encryption.

## Figures and Tables

**Figure 1 nanomaterials-13-00183-f001:**
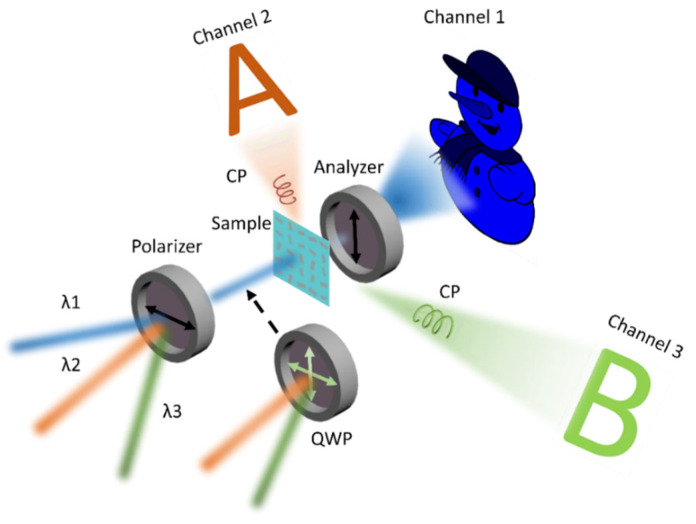
Operation schematic of three-channel metasurface for three wavelengths holography and nanoprinting. The metasurface is composed of TiO_2_ nanobricks with four kinds of structure and different orientation angles. The three images are displayed on three channels respectively. Channel 1 is a nanoprinting image obtained by an LP light that passes successively the metasurface and orthogonal optical path at a wavelength of λ_1_. Channel 2 and channel 3 are holographic images respectively obtained by CP light at two wavelengths. The polarizer is used to derive LP light for channel 1. A combination of a polarizer and a quarter-wave plate is used to derive CP light for channel 2 and channel 3.

**Figure 2 nanomaterials-13-00183-f002:**
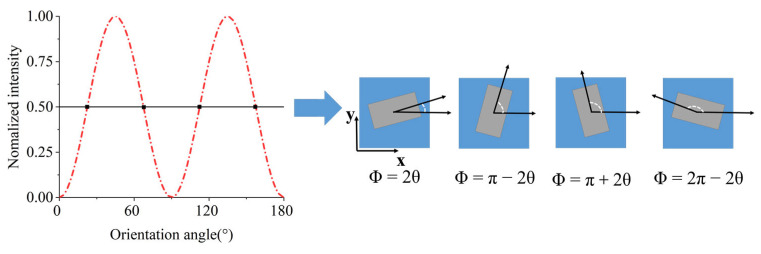
The principle of orientation angle degeneracy. The same output light intensity has four candidate orientation angles. The four candidate orientation angles can generate four different geometric phases.

**Figure 3 nanomaterials-13-00183-f003:**
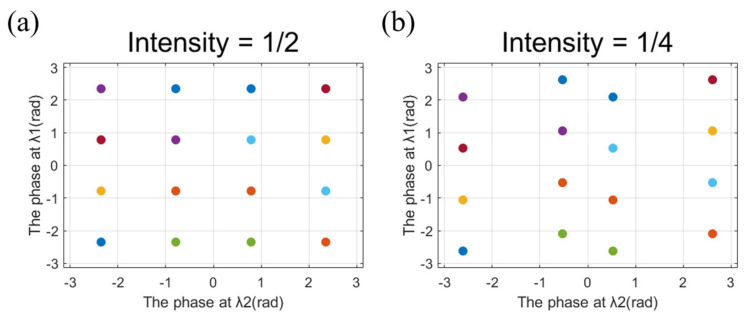
(**a**) The candidate phase distribution in 1/2 intensity. (**b**) The candidate phase distribution in 1/4 intensity.

**Figure 4 nanomaterials-13-00183-f004:**
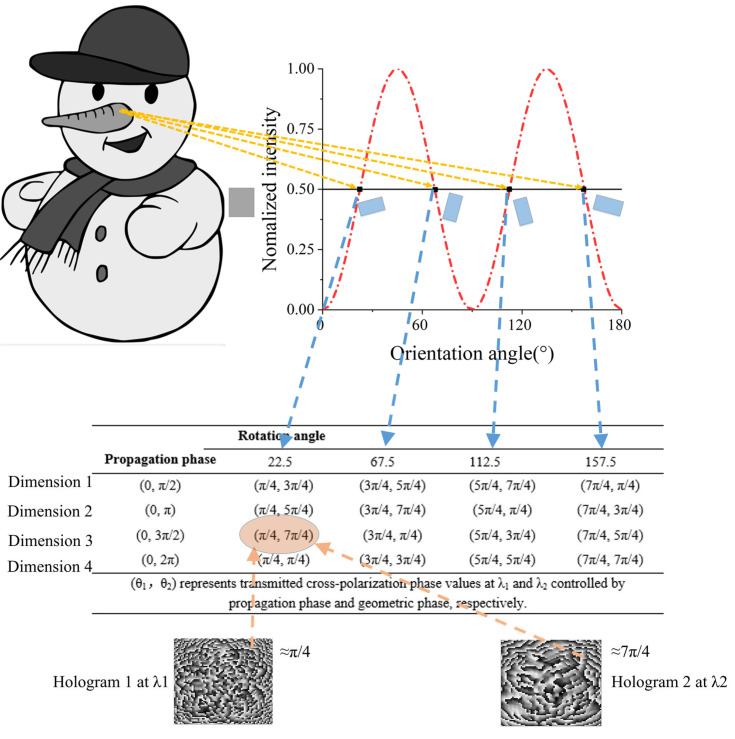
Principle schematic of the three-channel metasurfaces for multi-wavelength holography and nanoprinting.

**Figure 5 nanomaterials-13-00183-f005:**
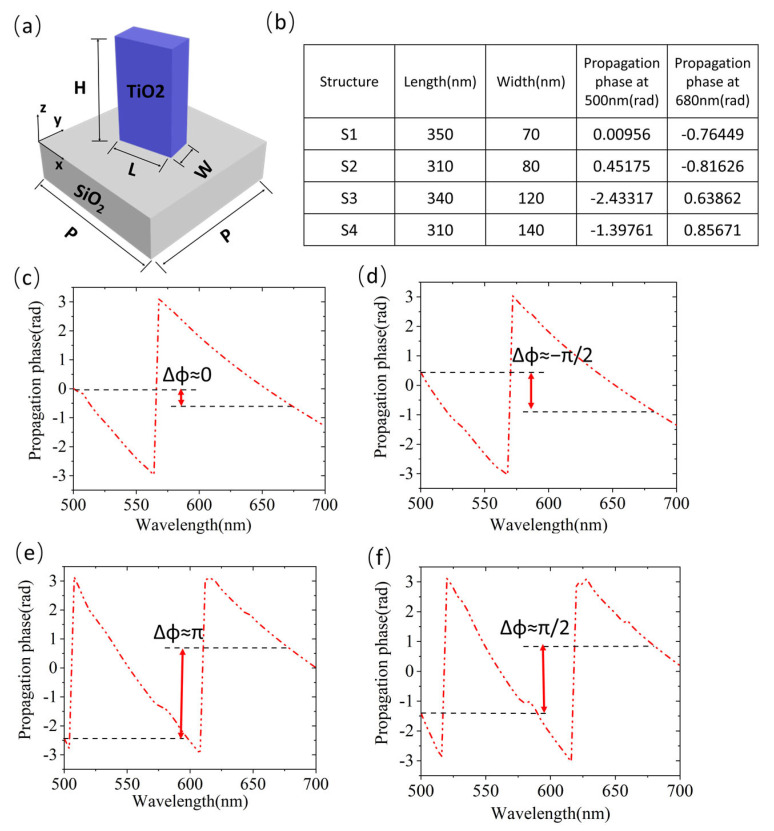
(**a**) Schematic of a nanostructure with a TiO_2_ nanobrick and a SiO_2_ substrate. (**b**) Each dimension parameter and propagation phase of the nanobrick. (**c**–**f**) Propagation phase curves of the four unit structures in the range of 500 nm to 700 nm. The red arrow and black dotted line show the phase difference between 680 nm and 500 nm in each dimension of the nanobrick.

**Figure 6 nanomaterials-13-00183-f006:**
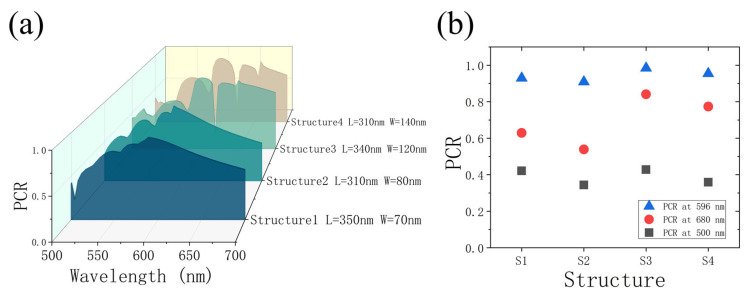
(**a**) Polarization conversion rate of the selected nanobricks under CP light ranging from 500 nm to 700 nm. (**b**) Polarization conversion rate of the nanobricks at 596, 680, and 500 nm respectively.

**Figure 7 nanomaterials-13-00183-f007:**
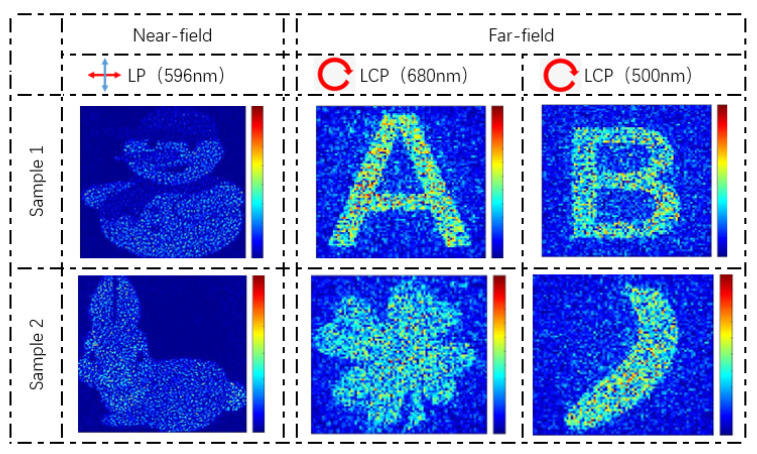
Simulation results of the metasurfaces for three-channel holography and nanoprinting. The first row and the second row respectively show the simulation results in sample 1 and sample 2. The first column shows the nanoprinting images under LP light (596 nm) in the near-field. The second column and the third column show the holographic images under CP light (680 nm, 500 nm) in the far-field. The red arrow represents the polarization direction α_1_ of the incidence light and the blue arrow represents the analyzer polarization α_2_.

**Figure 8 nanomaterials-13-00183-f008:**
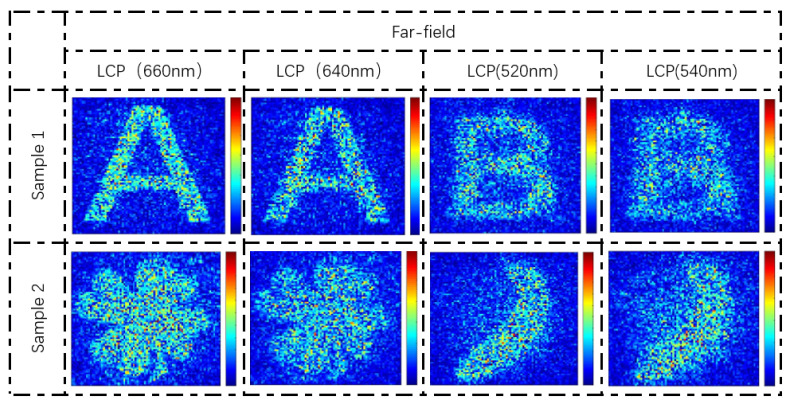
The simulation results of broadband response. The results include the holographic images by illuminating the metasurface under LCP light at different wavelengths (660 nm, 640 nm, 520 nm, 540 nm).

**Figure 9 nanomaterials-13-00183-f009:**
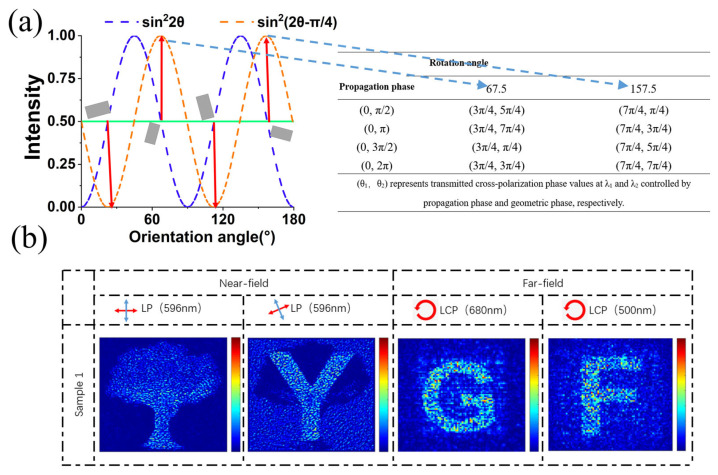
(**a**) Schematic diagram of the binary image encoding besides the three channels. Take the gray value of 0.5 as an example, four candidate orientation angles are obtained. If the corresponding pixel in the binary image is high level, the orientation of 67.5° and 157.5° can be chosen and 4 × 2 phase groups are listed. Select the most suitable phase group to realize dual-wavelength holography. (**b**) The simulation results of the four-channel holographic and nanoprinting images.

**Figure 10 nanomaterials-13-00183-f010:**
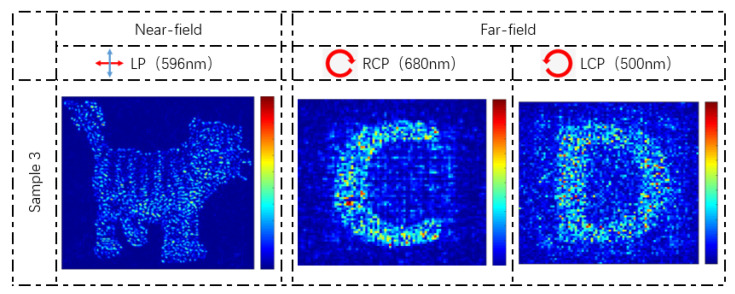
The simulation results of the holographic images under RCP at 680 nm and LCP at 500 nm.

**Table 1 nanomaterials-13-00183-t001:** The phase groups in 1/2 intensity.

	Rotation Angle			
**Propagation Phase**	22.5	67.5	112.5	157.5
(0, π/2)	(π/4, 3π/4)	(3π/4, 5π/4)	(5π/4, 7π/4)	(7π/4, π/4)
(0, π)	(π/4, 5π/4)	(3π/4, 7π/4)	(5π/4, π/4)	(7π/4, 3π/4)
(0, 3π/2)	(π/4, 7π/4)	(3π/4, π/4)	(5π/4, 3π/4)	(7π/4, 5π/4)
(0, 2π)	(π/4, π/4)	(3π/4, 3π/4)	(5π/4, 5π/4)	(7π/4, 7π/4)

(θ1, θ2) represents transmitted cross-polarization phase values at λ1 and λ2 controlled by propagation phase and geometric phase, respectively.

**Table 2 nanomaterials-13-00183-t002:** The phase groups in 1/2 intensity under LCP light and RCP light.

		Rotation Angle			
		22.5	67.5	112.5	157.5
	Geometric Phase	(π/4, −π/4)	(3π/4, −3π/4)	(5π/4, −5π/4)	(7π/4, −7π/4)
Propagation Phase					
(0, π/2)		(π/4, 3π/4)	(3π/4, 5π/4)	(5π/4, 7π/4)	(7π/4, π/4)
(0, π)		(π/4, 5π/4)	(3π/4, 7π/4)	(5π/4, π/4)	(7π/4, 3π/4)
(0, 3π/2)		(π/4, 7π/4)	(3π/4, π/4)	(5π/4, 3π/4)	(7π/4, 5π/4)
(0, 2π)		(π/4, π/4)	(3π/4, 3π/4)	(5π/4, 5π/4)	(7π/4, 7π/4)

(θ1, θ2) represents transmitted cross-polarization phase of RCP and LCP values at λ = 500 nm and λ = 680 nm, respectively.

## Data Availability

Not applicable.

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
