# Peer review of "Three-Channel Metasurfaces for Multi-Wavelength Holography and Nanoprinting"

_nanomaterials, 2022, doi:10.3390/nano13010183_

Round 1
Reviewer 1 Report
The authors of this theoretical paper present the design principle and numerical illustrative examples of specific multifunctional optical metasurfaces producing a pair of far-field holographic images and/or a near-field nanoimprinted image upon illumination by light of different polarizations and wavelengths.
The topic of multifunctional optical metasurfaces experiences a real surge of interest. Very different designs combining unusual performances in a single flat-optical device are being actively investigated. Particular feature of the design reported here is the possibility to use three different wavelengths for each function. This proposed principle is relatively fresh (or, at least, I am unaware of close analogs), and it nicely expands our knowledge of feasible metasurface devices and ways of their optimization. I consider the paper as generally publishable, after the authors fix a few issues listed below.
Major point:
Although the main idea is somewhat clear, it is impossible to evaluate its impact based on the provided data. In particular, it is necessary to address (with additional supporting data) the following questions:
1. How different (or how close) do the wavelengths need to be for the design procedure to be successful?
2. Why is the nanoimprinting limited to 4 shades and the holography to black/white images? Is it possible to achieve more levels and at what cost?
3. How does the achieved (poor) holographic image efficiency (about 50% and even 25%) compare to the literature data? Does the proposed concept allow for its improvement?
Minor:
- Eq. (2) is false. It should contain |T_L-T_S| instead of (T_L-T_S).
- Fig. 6 caption mentions nonexistent (c) and (d). All data is already in (b).
Reviewer 2 Report
This is an interesting work; Nevertheless some revision are needed in order to publish this work to MDPI Nanomaterials
1) Some minor syntax errors should be corrected, e.g. IO to become Io, TiO2 to become TiO2, etc.
2) I kindly ask the authors to discuss more why they chose the selected materials (TiO2, SiO2 etc). I would expect a discussion about the properties of those materials. Some more references would be helpful.
3) Could the authors discuss more about the holographic imaging in far field?
4) Could the authors present an SEM picture of their structures? The schematic in Fig. 5 is not enough. Have they fabricated such samples?
5) could the authors discuss more about the experimental properties of such samples? This work lacks of experimental data and information. I am afraid if the authors want to publish in MDPI Nanomaterilas some experimental data are necessary. Could they fabricate samples?
Could they compare their findings with experimental ones?
Reviewer 3 Report
In this manuscript, “Three-channel metasurfaces for multi-wavelength holography and nanoprinting,” the authors propose an efficient phase method to decouple wavelength and realize three-channel display operating at different wavelengths. Based on the obtained results, the authors claimed that the proposed approach may extend broad applications in information encryption, security, and color display. Overall, this manuscript has a strong potential for another review round after applying the issues and addressing the shortcomings listed below:
1-The authors should polish/revise some grammatical mistakes and typos along the manuscript. I invite the authors to read their manuscript carefully and make the required changes where necessary.
2-Please increase size of the text provided in the figures, where necessary.
3-In the Introduction section, while discussing recent developments in the field of metasurfaces, the following work should also be considered and cited to give a more general view to the possible readers of the work: [(i) Towards scalable plasmonic Fano-resonant metasurfaces for colorimetric sensing, Nanotechnology 33, 405201 (2022)].
4-What was the reason for using TiO2? Can we use any other composition? Please explain.
5-Corresponding references of the equations should be provided.
6-Please provide more detailed information regarding the simulation tool that is used to extract the simulation results.
Round 2
Reviewer 1 Report
In my first report, I have indicated the necessity to clarify the limits and capabilities of the proposed metasurface design concept. In their reply, the authors provide detailed explanations of:
- how different can be the operation wavelengths;
- why variously encoded images possess the particular color resolutions;
- how the achieved efficiency is compared with that of alternative metasurfaces from the literature.
It was absolutely puzzling for me, however, that none of this has been included into the paper. All these are important points. Clarifying them will help the reader and significantly improve the potential paper impact.
Also, I have to beg the authors’ pardon for a typo: I meant to mention the error in Eq. (1), which should contain the absolute value.
Reviewer 2 Report
The authors have revised their manuscript following most of my comments.
I feel this work could be published in its present form.
Reviewer 3 Report
Suitable for publication.
